# A Multi-Omics Study of Epigenetic Changes in Type II Alveolar Cells of A/J Mice Exposed to Environmental Tobacco Smoke

**DOI:** 10.3390/ijms25179365

**Published:** 2024-08-29

**Authors:** Qiyuan Han, Jenna Fernandez, Andrew T. Rajczewski, Thomas J. Y. Kono, Nicholas A. Weirath, Abdur Rahim, Alexander S. Lee, Donna Seabloom, Natalia Y. Tretyakova

**Affiliations:** 1Department of Biochemistry, Biophysics and Molecular Biology, University of Minnesota-Twin Cities, Minneapolis, MN 55455, USA; hanxx963@umn.edu (Q.H.); rajcz001@umn.edu (A.T.R.); 2Department of Medicinal Chemistry, College of Pharmacy, University of Minnesota-Twin Cities, Minneapolis, MN 55455, USA; fernandez.jenna@mayo.edu (J.F.); weira008@umn.edu (N.A.W.); rahim032@umn.edu (A.R.); 3Minnesota Supercomputing Institute, University of Minnesota-Twin Cities, Minneapolis, MN 55455, USA; konox006@umn.edu; 4Department of Biochemistry and Molecular Genetics, Feinberg School of Medicine, Northwestern University, Evanston, IL 60611, USA; alexanderlee2025@u.northwestern.edu; 5AeroCore Testing Services, University of Minnesota-Twin Cities, Minneapolis, MN 55455, USA; hartl008@umn.edu

**Keywords:** smoking, epigenetic changes, multi-omics, multi-omics analysis, type II cells, lung, methylation, hydroxymethylation, RNA-seq, proteomics, cancer

## Abstract

Lung cancer remains a major contributor to cancer fatalities, with cigarette smoking known to be responsible for up to 80% of cases. Based on the ability of cigarette smoke to induce inflammation in the lungs and increased lung cancer incidence in smokers with inflammatory conditions such as COPD, we hypothesized that inflammation plays an important role in the carcinogenicity of cigarette smoke. To test this hypothesis, we performed multi-omic analyses of Type II pneumocytes of A/J mice exposed to cigarette smoke for various time periods. We found that cigarette smoke exposure resulted in significant changes in DNA methylation and hydroxymethylation, gene expression patterns, and protein abundance that were partially reversible and contributed to an inflammatory and potentially oncogenic phenotype.

## 1. Introduction

Lung cancer is the leading cause of cancer death globally, resulting in more than 1.5 million deaths each year [1]. Smoking tobacco is a major risk factor for the development of lung cancer, with 90% of lung cancer cases in the United States occurring in individuals with a history of exposure to tobacco smoke [2]. Chronic inflammation of the lung, such as in chronic obstructive pulmonary disease (COPD), is known to increase the risk for lung cancer in smokers [3].

Although the exact mechanism by which inflammation contributes to lung cancer development is unknown, epigenetic deregulation may provide a possible link. Epigenetic changes such as the inactivation of tumor suppressor genes through promoter methylation and the activation of proto-oncogenes through promoter demethylation are recognized as a hallmark of lung cancer [4]. Importantly, such epigenetic changes appear at early stages of carcinogenesis and could be triggered by inflammation, which is known to generate pronounced changes in gene methylation and histone marks [5,6,7,8,9].

Tobacco smoke contains many organic and inorganic chemicals that may contribute to its inflammatory effects [9], such as nicotine, carbon monoxide, and lipopolysaccharides (LPS) [10]. Due to their potent inflammatory properties, inhaled LPS have been used to model COPD in mouse models [11,12]. LPS treatment increases the size and the multiplicity of lung tumors induced by tobacco-specific nitrosamine NNK [8]. Furthermore, recent studies have shown that the short-term inhalation exposure of laboratory mice to LPS leads to reduced levels of global cytosine hydroxymethylation and histone acetylation [6,7], as well as the increased methylation of tumor suppressor genes *Dapk1*, *Cdh13*, *Rassf1*, and *Tet1* and corresponding reduced gene expression patterns [7]. Collectively, these results indicate that the inflammatory reagent LPS present in cigarette smoke cause chronic inflammation in the lung, which could trigger early epigenetic changes contributing to lung cancer development. However, there is a lack of systematic studies on the timing and reversibility of the smoking-associated inflammation and epigenetic dysregulation.

In the present study, we employed a multi-omic approach to characterize the timing and the reversibility of the epigenetic effects of cigarette smoke exposure in target cells for lung cancer development (Figure 1A). As subtle changes in the transcriptome or proteome of specific cell types can be masked in the analysis of bulk tissues, we specifically characterized epigenetic changes in Type II pneumocytes, which are known to be the target cells for adenocarcinomas stemming from cigarette smoke exposure [13,14]. Type II pneumocytes are endothelial cells that, along with Type I pneumocytes and capillary endothelial cells, constitute alveoli, where the lungs and the blood exchange oxygen and carbon dioxide during the process of breathing in and breathing out. We have conducted a detailed multi-omic investigation integrating epigenomics, transcriptomics, and proteomics to determine the molecular mechanisms contributing to oncogenesis induced by cigarette [smoke exposure (Figure 1B).

## 2. Results

### 2.1. Animal Studies

Our aim was to characterize epigenetic changes induced by exposure to environmental cigarette smoke (ECS) in Type II alveolar lung cells. Fourteen A/J mice of each sex were exposed to cigarette smoke starting at 6 weeks of age for either 3 or 10 weeks (Figure 1A). A subset of mice was treated for 10 weeks and then allowed to recover for 4 weeks. Mice were subjected to whole body exposure to cigarette smoke for 4 h per day, 5 days per week. Within the exposure chamber, the levels of CO (226.90 ± 36.37 ppm) and the total particulate matter (TPM) (95.01 ± 16.49 mg/m^3^) were monitored throughout the exposures. The observed mortality rate for the 10-week exposure was 8.9% over the full 10 weeks, with decreases in the survival probability observed at 7 and 9 weeks (Appendix A). This was the only exposure observed to cause fatalities. Body weight increased throughout the study in all groups. However, a diminished weight gain occurred in mice exposed to cigarette smoke for 10 weeks as compared with the control mice (female control: 19.41 ± 1.19 g, female smoke: 16.86 ± 0.62 g, *p* = 7.02 × 10^−9^; male control: 24.68 ± 2.49 g, male smoke: 21.34 ± 1.37 g, *p* = 2.02 × 10^−5^) (Appendix A). Post-exposure effects, such as lethargy, tremors, and a hunched posture, were reported multiple times in most ECS-exposed mice throughout the exposure period.

### 2.2. Histopathological Examination of Lung Tissues

Whole lungs from male and female A/J mice in 10 weeks of ECS exposure w/wo 4 weeks of recovery were subjected to histopathological analysis. The lungs of mice exposed to ECS for a total of 10 weeks (with or without 4 weeks of recovery) demonstrated a low degree of an inflammatory response (Appendix A). The most observed changes associated with exposure to ECS were an increased number of alveolar macrophages, many of which had a pale green-to-brown granular cytoplasmic pigment, and multifocal vascular congestion (Appendix A).

### 2.3. Global Changes in DNA Methylation and Hydroxymethylation in Type II Alveolar Cells of Mice Exposed to ECS

A quantitative mass spectrometry-based approach previously reported by our group [6,7] was used to quantify global cytosine methylation and hydroxymethylation in genomic DNA isolated from Type II alveolar lung cells of A/J mice exposed to ECS, as well as matched controls exposed to filtered air (Figure 2A). Global cytosine methylation levels were essentially unchanged in the 3-week, 10-week, and post-exposure groups except for a small increase in male mice from 3-week exposure group (Figure 2B, top panel). For the global levels of 5 hmC, we also observed a gender-specific response to ECS. In female mice, global levels of 5 hmC were unchanged in the 3-week and 10-week exposures; however, we did observe an increase in levels of 5 hmC in the post-exposure group (female post-exposure control: 0.18 ± 0.01, female post-exposure smoke: 0.21 ± 0.01% dC, *p* = 0.0129) (Figure 2B, bottom panel). In male mice exposed to ECS, 5 hmC levels were significantly increased in the 3-week exposure group, but remained unchanged in the other treatments (Figure 2B, bottom panel). No global methylation or hydroxymethylation changes were observed in the genomic DNA isolated from liver tissues of the same animals (Appendix A). Overall, these results suggest that there is a sex- and tissue-specific epigenetic response to ECS in Type II alveolar cells of A/J mice. Interestingly, female smokers are known to be at a higher risk of lung cancer as compared to men [15]. Furthermore, women have also been shown to have a significantly lower level of DNA methylation than men [16]. Therefore, known epigenetic differences between males and females may play a role in response to cigarette smoke exposure.

### 2.4. Differential Methylation Analysis Reveals Extensive DMRs and DhMRs in Type II Alveolar Epithelial CELLS of Female A/J Mice Exposed to ECS

While global analyses of cytosine methylation and hydroxymethylation revealed modest overall changes in DNA epigenetic marks in Type II cells upon exposure to ECS (Figure 2B), they cannot determine where such changes are taking place. To probe for site-specific methylation and hydroxymethylation changes in DNA induced by ECS, we conducted reduced representation bisulfite sequencing (RRBS) combined with oxidative-RRBS (oxRRBS) sequencing, which allows a distinction between 5 mC and 5 hmC [17]. Genomic DNA isolated from Type II alveolar cells of female A/J mice (3-week/10-week/post-exposure control and ECS groups, *n* = 3 per group, each sample was pooled from three mice) was prepared for sequencing using Tecan Ovation RRBS Methyl-Seq with TrueMethyl oxBS modules [17]. We focused on epigenetic changes in genomic DNA of females, since female smokers are at a higher risk for lung cancer development [15].

Our analyses specifically characterized symmetrically methylated CpG sequences, which is the main pattern of cytosine methylation within the genome [18]. Within the human genome, 70–80% of CpG sites are fully methylated [19], and specialized DNA methyltransferase DNMT1 specifically targets hemimethylated CpG dinucleotides generated during DNA replication. After filtering for symmetric methylation, blacklisted regions, and the sequencing depth, a total of 943,506, 1,134,594, and 818,932 CpG sites were covered by both RRBS and oxRRBS data sets in 3-week, 10-week, and post-exposure groups, respectively. In all three treatment groups, 706,733 of the CpG sites are covered. Since DNA methylation is thought to exert its function in sequences containing multiple CpG sites rather than in isolated CpGs [20], we further processed the data to identify differentially methylated regions (DMRs) and differentially hydroxymethylated regions (DhMRs). DMR/DhMR is defined as a genomic region that contains at least three CpG sites within a 200 bp genomic window, with a false discovery rate of less than 0.05 [21].

Overall, many more DhMRs were identified in each treatment group than DMRs, indicative of a more dynamic response of DNA hydroxymethylation to the ECS exposure (Figure 3A,B). Specifically, in the 3-week ECS exposure group, we observed a total of 12 DMRs (Figure 3A) and 1823 DhMRs (Figure 3B). Among the DMRs, 11 were hypermethylated and one was hypomethylated following exposure to ESC. Among DhMRs, 961 had increased hydroxymethylation and 862 had decreased hydroxymethylation levels. For the 10-week ECS exposure group, we observed a total of 17 DMRs, with two being hypermethylated in the smoking group and 15 being hypomethylated upon smoking exposure (Figure 3A). From the total of 3425 DhMRs (Figure 3B) observed in the 10-week ETS exposure group, 1926 had increased hydroxymethylation and 1499 exhibited decreased hydroxymethylation. Interestingly, Type II cells of mice in the post-exposure group had a greater number of DMRs than those of animals sacrificed immediately after ECS treatment, indicating that inhalation exposure to ECS has a delayed effect on the methylome (Figure 3A). A total of 345 DMRs were observed in the post-exposure group, of which 341 were hypermethylated and 4 were hypomethylated (Figure 3A). Among 1776 DhMRs identified in the ECS post-exposure group, 507 were hyper-hydroxymethylated and 1269 were hypo-hydroxymethylated (Figure 3B).

The observed DMRs and DhMRs were widely distributed across different genomic features (Figure 3C). In the 3-week ECS exposure group, 16.7% of DMRs and 29.7% of DhMRs were found in promoter regions, 58.3% of DMRs and 33.5% of DhMRs were identified in the gene bodies, while 25.0% of DMRs and 29.4% of DhMRs were in distal intergenic regions. In the 10-week ECS exposure group, 29.41% of DMRs and 33.6% of DhMRs were found in the promoter regions, 47.1% of DMRs and 32.73% of DhMRs were identified in the gene bodies, and 17.6% of DMRs and 27.8% of DhMRs were in distal intergenic regions. In the post-exposure group, 40.3% of DMRs and 46.9% of DhMRs were localized in promoter regions, 38.3% of DMRs and 31.2% of DhMRs were identified in the gene body, and 15.1% of DMRs and 14.4% of DhMRs were found in distal intergenic regions.

As shown in Figure 3A,B, Type II alveolar cells of A/J mice exhibited many more differentially hydroxymethylated regions after exposure to ECS as compared to differentially methylated regions. Therefore, additional analyses were conducted to characterize DNA hydroxymethylation differences across treatment groups. To investigate the reversibility of DNA hydroxymethylation changes, we compared DMRs and DhMRs observed in mice immediately sacrificed after exposure to ECS to the group that was allowed to recover for 4 weeks (post-exposure group). Each of the ECS treatment groups had several unique DhMR, but 165 DhMRs were shared among all three treatment groups (Appendix A). To identify the biological pathways enriched among DhMR-containing genes across the different treatment groups, KEGG pathway enrichment analysis was conducted (Figure 3D). We found that the MAPK signaling pathway was significantly enriched among DhMR containing genes in all three treatment groups (Figure 3D). The MAPK signaling pathway exerts a critical role in bridging the extracellular signals and intracellular responses, which is associated with cell proliferation, differentiation, and inflammation [22,23,24]. Alterations of this pathway are found in many cancers including lung cancer due to genetic changes and epigenetic dysregulation [25,26]. In addition, the Rap1 signaling pathway and the hippo signaling pathway were enriched in both the 3- and 10-week ECS treatment groups (Figure 3D). The Rap1 signaling pathway is involved in many basic cellular functions including cell adhesion and cell–cell junctions and was found to be a key factor in regulating cell invasion and metastasis in cancer [27]. The hippo signaling pathway plays a pivotal role in modulating cell proliferation and has been shown to be involved in the initiation and development of lung cancer [28]. The Wnt signaling pathway, which plays an important role in lung carcinogenesis [29], was significantly enriched in the post-exposure group (Figure 3D). The PI3k-Akt signaling pathway, which regulates the cell proliferation, apoptosis, metastasis, and EMT of lung cancer [30], was enriched in the 3-week ECS group (Figure 3D). Apart from that, signaling pathways regulating the pluripotency of stem cells, including the MAPK signaling pathway, Wnt signaling pathway, PI3K-Akt signaling pathway, as well as the TGFB signaling pathway, were significantly enriched in both the 10-week and post-exposure groups (Figure 3D). Therefore, exposure to ECS induces cytosine hydroxymethylation changes in multiple genes and influences biological pathways involved in lung carcinogenesis; these epigenetic changes could contribute to the development of smoking-induced lung cancer. Lastly, the nicotine addiction pathway was enriched in the 10-week ECS group, suggesting that changes in DNA hydroxymethylation could contribute to the development of nicotine addiction (Figure 3D).

### 2.5. Gene Expression Changes in Type II Alveolar Epithelial cells of A/J Mice Exposed to ECS

RNA-Seq analyses were conducted to identify changes in gene expression in Type II alveolar epithelial cells of A/J mice treated with ECS. RNA-seq results for the top 500 variant genes clustered in accordance with their treatment group (Figure 4A). Specifically, ECS treatment for 3 and 10 weeks induced striking gene expression changes, whereas post-exposure groups clustered together with the air control samples, indicating that most gene expression changes induced by ECS were reversible (Figure 4A,B).

Exposure to cigarette smoke altered the expression of numerous genes in murine Type II pneumocytes. Specifically, there were a total of 304, 209, and 1 differentially expressed genes (DEG) in the 3-week ECS, 10-week ECS, and post-exposure groups, respectively. In addition to some unique DEGs in the 3- and 10-week ECS exposure, a significant overlap of DEGs between the 3- and 10-week exposure groups was also observed (Appendix A). The only DEG in the post-exposure group was *Fosb*, which was also significantly upregulated in the 3-week ECS group but not in the 10-week group (Appendix A). In humans, this gene encodes a protein that can dimerize with JUN proteins to form transcriptional complex AP-1, which has been implicated as a oncogenic or anti-oncogenic regulator in cell proliferation, differentiation, apoptosis, and tumor invasion in various cancer types [31].

In order to identify the patterns of gene expression changes with an increasing exposure time to ECS and post-exposure, we conducted gene clustering analyses of DEGs that have been identified in different ECS treatment groups. Following a filtering step in RNA-seq data analysis to eliminate genes with non-significant changes, a total of 373 DEGs were detected in all three groups. We identified two major patterns of gene expression changes, as shown in Figure 4C. In Cluster 1, 139 genes were downregulated following exposure to ECS for 3 weeks and further downregulated in the 10-week exposure group, followed by a return to base levels in the post-exposure group. These genes are enriched in KEGG pathways such as transcriptional regulation and cell cycle progression (Figure 4D, top panel). These included *Ccna2*, *H3c15*, *H3c2*, *H3c3*, *H3c4*, and *Mycn*. *Ccna2* encodes the Cyclin A2 protein which regulates the progression through the cell cycle. The increased expression of Ccna2 has been documented in various tumor types including lung cancer, and it has been proposed to be a biomarker for cancer diagnosis [32,33,34]. *H3c15*, *H3c2*, *H3c3*, and *H3c4* are all members of the histone H3 family, which are likely to be involved in chromatin packaging and gene regulation. *Mycn* is an oncogenic driver in many types of cancer including lung cancer [35]. DEGs belonging to the cell cycle pathway include *Ccna2*, *Ccnb1*, *Ccnb2*, *Cdk1*, *Espl1*, *Pkmyt1*, *Plk1*, and *Ttk*. *Ccnb1*, *Ccnb2*, and *Cdk1* are all members of the mature-promoting factor (MPF), which is a key regulator in inducing the G2/M transition [36]. The overexpression of *Espl1* has been reported to be associated with augmented malignancy in lung cancer [37]. *Pkmyt1*, *Plk1*, and *Ttk* are upregulated in lung cancer and are considered oncogenes due to their function in promoting cell proliferation [38,39,40].

In contrast, 234 genes in Cluster 2 showed an opposite trend as compared to Cluster 1, with an upregulation in the 3-week and 10-week exposure groups and a return to base levels in the post-exposure groups. These genes are enriched in the circadian rhythm pathway and include *Bhlhe41*, *Nr1d1*, *Per1*, *Per2*, and *Per3* (Figure 4E bottom panel). Previous work has shown that exposure to environmental cigarette smoke disrupts the circadian clock function in mice [41]. This disruption has been shown to cause a downregulation in the expression of *Arntl* (*BMAL1*), a core circadian clock gene, and leads to lung inflammation and injury through the increased release of the proinflammatory cytokines CCL1 and CXCL [41].

### 2.6. Protein Abundance Changes in Type II Alveolar Epithelial Cells of A/J Mice Exposed to ECS

Changes in protein abundances in type II alveolar epithelial cells were determined using LC-MS-based bottom-up proteomics methodologies to elucidate the explicit phenotypic response to ECS exposure with and without post-exposure recovery. Preliminary analyses of the proteome of control and exposed mice where the two sexes were combined (*n* = 6) showed no significant changes in the protein abundance in response to ECS. Therefore, samples were separated by sex (*n* = 3 each). As before, we focused on protein abundance changes in female mice as female smokers are at a higher risk for lung cancer development [34].

Volcano plots detailing the differential abundance of proteins with ECS exposure in female mice (*n* = 3) are presented in Figure 5. After three weeks of ECS exposure, the levels of Atp1b1, Ppp2ca, Letm1, and Ctsd proteins were increased in abundance, while Ccdc6 showed a decrease in abundance (Figure 5A). After 10 weeks of ECS exposure, 10 proteins increased in abundance in the Type II pneumocytes of female mice (Figure 5A). A gene ontology (GO) analysis of proteins upregulated in ECS-exposed mice identified a variety of pathways including ascorbic acid biosynthesis and the negative regulation of ubiquitin-specific protease activity, among others. (Figure 5B). The genes Fubp1 [42], Cox6b1 [43], Pon3 [44], Iars2 [45], Ctsd [46], and Akr1b1 [47] have been previously shown to promote lung cancer proliferation.

In Type II pneumocytes of female mice that were allowed to recover for 4 weeks following 10 weeks of ECS exposure there were 24 proteins that showed significantly increased abundance and 1 protein that showed a significantly decreased abundance (Figure 5A); of these 24 proteins, 2- Pon3 and Ctsd- were initially noted to be increased in abundance after 10 weeks of ECS exposure and remained increased in abundance after the four-week recovery. A GO analysis of those proteins in female Type II pneumocytes with increased abundance after 10 weeks of ECS exposure and 4 weeks of recovery indicates the enrichment of pathways associated with protein expression and secretion, suggesting an increase in signaling following recovery from exposure in Type II pneumocytes of female mice.

Male mice were found to have 24 differentially abundant proteins after 10 weeks of ECS exposure and 76 differentially abundant proteins after 10 weeks of ECS exposure with 4 subsequent weeks of recovery (Appendix A). Interestingly, no proteins with differential abundance were shared between female and male mice. In addition, no genes which showed a significant change at the mRNA level were observed to have significantly altered protein abundances in any animals tested.

### 2.7. Integration of Epigenomic and Transcriptomic Data Identifies DEGs Regulated by DNA Methylation and Hydroxymethylation

To identify potential epigenetic drivers of lung cancer development, further analyses were conducted focusing on the genes that exhibited changes in DNA methylation/hydroxymethylation and also demonstrated differential levels of mRNA expression (Figure 6). These analyses identified several cancer-related genes, including *Kif26a*, *Acer2*, *Serpine1*, *Nr1d2*, *Efnb2*, and *Chst11*, to be affected by ECS exposure. In type II pneumocytes of mice exposed to cigarette smoke for 3 weeks, *Kif26a* expression was increased, which was accompanied by a decrease in cytosine hydroxymethylation in the distal intergenic region (Figure 6A). *Kif26a* has been shown to promote cell proliferation and G0/G1 phase cell cycle progression in breast cancer cells [48]. *Acer2* was upregulated and showed decreased CpG hydroxymethylation levels in the coding region of the gene following 3 weeks of treatment with ECS (Figure 6A). *Acer2* has been shown to promote tumor growth and angiogenesis via catalyzing the formation of sphingosine-1-phosphate precursor sphingosine [49]. Increased expression levels and increased cytosine hydroxymethylation levels were observed for the *Serpine1* gene in the type II cells of mice exposed to cigarette smoke for 3 weeks (Figure 6A). The overexpression of *Serpine1* can lead to increased tumor proliferation and tumor budding while inhibiting apoptosis in primary tumors [50].

For animals exposed to ESC for 10 weeks, *Nr1d2* was upregulated and showed decreased cytosine hydroxymethylation levels in the promoter region of the gene (Figure 6B). *Nr1d2* encodes a transcriptional repressor that plays a role in regulating circadian rhythms [51]. A recent study reported that NR1D2 promotes the proliferation and mobility of glioblastoma cells [51]. In that study, AXL was identified as a new transcriptional target of NR1D2: the regulatory effect of NR1D2 on the PI3K/AKT axis promotes the proliferation, migration, and invasion of glioblastoma cells [51]. The increased expression and decreased levels of cytosine hydroxymethylation in introns of both *Efbn2* and *Chst11* genes were characteristic for Type II cells of mice exposed to cigarette smoke for 10 weeks as compared to the controls (Figure 6B). It has been reported that targeting the Efbn2 protein with a highly specific antibody inhibits angiogenesis and tumor growth, suggesting an oncogenic role for this gene [52]. Previous research has suggested that *Chst11* promotes lung cancer metastasis via the changing intracellular iron metabolism [53]. In breast cancer, the expression of *Chstl11* is controlled by DNA methylation [54]. Therefore, the epigenetic deregulation and differential expression of *Chst11* may play a role in the initiation of smoking-induced lung cancer.

### 2.8. Identification of Smoking-Induced Early Epigenetic Changes Found in Lung Adenocarcinoma (LUAD)

To determine which of the smoking-induced early epigenetic changes identified in the A/J mouse model are relevant to human adenocarcinoma, we conducted further analyses using the methylation and RNA-seq dataset from the TCGA human lung adenocarcinoma project (the results shown here are in whole or part-based upon data generated by the TCGA Research Network: https://www.cancer.gov/tcga (accessed on 31 August 2021 (RNA-seq dataset) and on 24 November 2021 (DNA methylation dataset)) and compared our results to the human LUAD results.

The human LUAD methylation dataset was generated from the Illumina Infinium HumanMethylation 450 beadchip or HumanMethylation27 beadchip, which did not differentiate 5 hmC from 5 mC and is limited to preselected CpG sites. We only analyzed sites from the HumanMethylation450 beadchip to avoid a loss of sites due to a lack of overlap between the HumanMethylation450 beadchip and the HumanMethylation27 beadchip. By analyzing 473 primary tumor samples (TP) and 32 healthy tissue samples (NT), we identified a total of 67,559 differentially methylated sites (DMS) at a cutoff value < 0.05 and a methylation difference > 10% (Figure 7A). Among these DMS, 33,954 were hypermethylated in tumor samples and 33,605 were hypomethylated in tumor samples (Figure 7A). The genome coordination of these DMS were then lifted from the hg38 genome to the mm10 genome via the liftover function in the rtracklayer package [55]. After genome liftover, 47,700 DMS were found in homologous sites in the mm10 genome. To allow for a direct comparison, we also conducted DMS analysis for the RRBS and oxRRBS datasets in our mouse study. With a significance cut off at *p* < 0.05 and methylation difference > 10% of our mouse study, we identified a total of 58 sites that were differentially methylated in human LUAD while also being differentially methylated/hydroxymethylated in at least one of the ECS treatment groups. This number increased to 147 sites if we relaxed the threshold of significance to be a methylation difference greater than 5% Next, we analyzed the RNA-seq dataset of human LUAD from the Cancer Genome Atlas (TCGA) program. Between 533 primary lung adenocarcinoma tumors and 59 matching normal tissues, 2094 DEGs (with FDR < 0.05 and FC > 2) were identified. A further intersection of the 147 sites and DEGs in human LUAD identified 10 genes that showed an early epigenetic change upon ECS exposure, and this change persisted in LUAD and was associated with the differential expression of the corresponding genes (Figure 7C). These genes include *GRIA3*, *TAF7L*, *ARL14*, *PCDHGA9*, *MDGA1*, *ZFPM2*, *OSR2*, *TMC8/TMC6*, *HCN2*, and *CDK5R2*, which are likely to be relevant in human lung cancer and are also epigenetically deregulated upon exposure to ECS in our animal study.

In addition, siRNA knockdown was employed to probe the functional roles of ZFPM2 and OSR2 in lung cancer cell proliferation. We focused on these two genes because of their transcriptional factor function, which is a class of key enablers of cancer stemness. qRT-PCR analyses showed a 47% and 70% knockdown efficiency of ZFPM2 and OSR2, respectively (Appendix A). Importantly, the knockdown of OSR2 in human adenocarcinoma (H838) cells significantly inhibited cell proliferation as compared to the non-targeting siRNA group (Figure 7D), suggesting a possible oncogenic role of OSR2 in lung cancer. The knockdown of ZFPM2 in H838 cells slightly reduced cell proliferation [56], but these differences were not statistically significant (Figure 7D).

### 2.9. Identification of Smoking-Induced Protein Abundance Changes Relevant to Lung Adenocarcinoma (LUAD)

Publicly available proteomics datasets from the CPTAC3-Discovery lung adenocarcinoma (LUAD) study were downloaded from the Clinical Proteomic Tumor Analysis Consortium (CPTAC) [57] and run through the Perseus data analysis suite to reveal the proteins that were significantly increased and decreased in abundancein lung cancer. Due to the large number of samples in the cohort (117 tumor samples and 101 normal samples), many proteins were found to be significantly increased and decreased in these datasets (3578 proteins increased and 2789 proteins decreased).

Having identified ten gene products that were significantly increased in abundance after 10 weeks of cigarette smoke exposure with and without a post-exposure recovery period, we then compared the list of proteins increased in lung adenocarcinoma in the LUAD data with the list of proteins identified in our study to determine their potential involvement in smoking-induced lung cancer. This analyses identified four proteins—Pdhx, Psma6, Ruvbl1, and Ywhaq—that were increased in abundance as a result of ECS exposure and also amplified in human lung adenocarcinomas.

To ascertain the functional roles of Psma6, Ruvbl1, Pdhx, and Ywhaq in lung adenocarcinoma, siRNA knockdown of these genes was performed, followed by cell proliferation assay. We noted that the knockdown of Pdhx in human adenocarcinoma (H838) cells significantly inhibited cancer cell proliferation as compared to the non-targeting siRNA group (Appendix A), suggesting a possible oncogenic role of Pdhx in smoking-induced lung adenocarcinoma. With the knockdown of Psma6, Ruvbl1, and Ywhaq, H838 cells showed an initial reduced cell proliferation over the first four days of the assay, though by day 5, the growth rates were not significantly different from the control cells (Appendix A).

## 3. Discussion

Smoking is a major risk factor for lung cancer, increasing the chances for cancer development 30-fold in current smokers and 20-fold in former smokers [58,59]. The most common type of smoking-induced tumor is pulmonary adenocarcinoma, which originates in Type II epithelial cells of the lung [13,59]. Cigarette smoke contains at least 60 known carcinogens and dozens of pro-carcinogens, including inflammatory agents such as lipopolysaccharides (LPS) [60,61]. We have previously conducted preliminary studies to characterize gene expression changes in Type II cells of A/J mice treated with inflammatory agent LPS [6] and examined epigenetic changes in the bulk lung tissue of A/J mice exposed to cigarette smoke [7]. However, our previous studies have not specifically characterized epigenetic changes in Type II cells following exposure to cigarette smoke. Furthermore, the reversibility of these changes and their potential role in lung cancer development have not been previously addressed.

In the present work, male and female A/J mice were exposed to cigarette smoke (ECS) for 3 or 10 weeks and animals in the 10-week exposure group were sacrificed either immediately or following a 4-week recovery period. Alveolar Type II epithelial cells were isolated, and global and site-specific changes in methylation, hydroxymethylation, and gene expression were characterized in this specific cell type in different ECS treatment groups.

Global levels of DNA methylation were unchanged in all exposure groups, with the exception of a small increase in C-5 cytosine methylation observed in males exposed to ECS for 3 weeks (Figure 2B, top). Whole-body exposure to ECS for 3 or 10 weeks had a small effect on the global levels of hmC (Figure 2B, bottom). However, we observed a small increase in global hmC in genomic DNA isolated from Type II cells of male mice exposed to ECS for 3 weeks and in female mice exposed to cigarette smoke for 10-weeks and allowed to recover for 4-weeks (post-exposure group) (Figure 2B, bottom). Women are known to be at a higher risk of smoking-induced lung cancer than men [15]. Epigenetic differences between men and women have also been previously demonstrated [16], which may help explain the different lung cancer risk between men and women.

Our sequencing results revealed significant alterations in the DNA methylome and hydroxymethylome in mice exposed to cigarette smoke. RRBS/oxRRBS was employed to separately map DNA methylation and hydroxymethylation changes in type II cells isolated from mice exposed to cigarette smoke for 3 weeks, 10 weeks, or post-10 weeks. Overall, 5 hmC was significantly more affected than mC upon exposure to ECS (Figure 3A,B), whereas there was a delay in the DNA methylation response to ECS exposure (Figure 3A). Genes containing DhMR were enriched in cancer-related pathways, such as the MAPK signaling pathway, Hippo signaling pathway, and Wnt signaling pathway (Figure 3D), suggesting a potential role of 5 hmC in the development of smoking-induced lung cancer.

Type II pneumocytes of mice exposed to ECS for 3 or 10 weeks exhibited significant changes in the gene expression patterns. The greatest changes in gene expression were observed in genes involved with the circadian rhythm, glutathione metabolism, and xenobiotic metabolism. Specifically, we observed expression changes in multiple genes involved in the circadian rhythm such as *Dbp*, *Nrd1d1*, *Nrd1d2*, *Bhlhe41*, *Npas2*, and *Arntl* [62,63,64]. Recent studies revealed that inflammation and immune functions are governed, in part, by the circadian timing system [41]. Consistent with previous work, we observed a downregulation of *Arntl* [41]. *Arntl* is a core circadian clock gene that regulates various genes involved in the circadian rhythm and has also been shown to regulate inflammatory responses [65]. The downregulation of *Arntl* due to cigarette smoke exposure has been shown to be associated with an increase in lung inflammation and injury [41]. These results suggest that the deregulation of molecular clocks could be associated with lung inflammation and injury caused by exposure to cigarette smoke.

Changes in the protein abundance in response to ECS exposure were examined using LC-MS-based bottom-up proteomics. The analysis of protein abundances in the type II pneumocytes of female mice subjected to ECS for different periods of time showed distinct changes in the protein abundance at each time point, including the increased abundance of several oncoproteins after 10 weeks of exposure, where some of these changes persisted after 4 weeks of recovery. Importantly, male and female samples showed distinct differences in protein abundance responses to ECS, suggesting a sex-dependent response to cigarette smoke exposure, a phenomenon that has been observed in other studies [66]. In addition, we failed to identify any genes that showed significant changes at both at the transcriptomic and proteomics levels; this apparent discrepancy could be due to an insufficient depth of coverage in our proteomics experiments that employed small protein amounts. This could also be accounted for by the epigenetic regulation of protein expression via mechanisms like miRNA, which will also be investigated in future experiments.

To identify the epigenetic drivers of lung diseases associated with smoking, DNA methylation and hydroxymethylation data were integrated with the transcriptomic data. These analyses have identified several genes that contained DMRs or DhMRs and also demonstrated differential expression (Figure 7). This includes genes involved in cancer progression such as *Kif26a*, *Acer2*, *Serpine1*, *Nr1d2*, *Efnb2* as well as *Chst11*, which have all previously been shown to be deregulated in various forms of cancer, including lung cancer. The potential roles of these genes in smoking-induced lung cancer deserve further investigation as they could be potential treatment targets via epigenetic manipulation.

When comparing the molecular alterations in the Type II pneumocytes of A/J mice exposed to ECS to those seen in human lung adenocarcinoma, we identified multiple CpG sites that were differentially methylated and genes that were deregulated in both our mouse smoke inhalation study and in human lung adenocarcinoma. These overlaps highlight the CpG sites and genes that have an early diagnostic or therapeutic value in the early stage of smoking-induced lung cancer. Of these genes, we identified OSR2 as a potential oncogene which was epigenetically dysregulated upon ECS exposure in our mouse study as well as in human lung adenocarcinoma dataset and upregulated in the human lung adenocarcinoma. The knockdown of OSR2 in the lung tumor cell line led to decreased levels of cell proliferation, supporting its functional role. These results shed light on the additional pathways affected in smoking-induced lung cancer.

While these results represent an important look at the molecular mechanisms underlying the transition from inflammation to oncogenesis, there are also several important limitations to this study which future work will consider. The mice were only exposed for a maximum of 10 weeks which, while representing a significant portion of their lives, does not serve as a good proxy for the long-term smoking habits seen in many human cancer patients. Additionally, the mice with 10 weeks of treatment have an 8.9% mortality rate with the observation of vascular congestion and intra-alveolar hemorrhage, which might cause confounding effects in addition to inflammation-associated omics changes; future studies should consider longer-term exposures with a reduced dosage on each day. In addition, future studies should utilize larger sample sizes for each sex to capture subtle changes in the proteome that may have been missed in this study. Furthermore, this study has only investigated the omics changes in type II alveolar epithelial cells, whereas other types of cells such as pulmonary neuroendocrine cells, basal cells, and club cells could also be the origin of lung cancer or indirectly contribute to carcinogenesis through cell signaling [67]; future studies could harness single-cell technology to capture the full complexity of lung cancer development. Finally, future studies should utilize RNA-Seq techniques beyond mRNA to examine the miRNA epigenetic control of gene expression.

Overall, our comprehensive study has characterized changes in DNA methylation, hydroxymethylation, gene expression patterns, and protein abundances in Type II pneumocytes of A/J mice following inhalation exposure to ECS. We found that exposure to ECS leads to significant changes in the epigenome of murine Type II pneumocytes. This epigenetic deregulation, along with genetic mutations, likely contributes to the development of smoking-induced lung cancer. Furthermore, this work identified several cancer-related genes which demonstrate epigenetic deregulation along with differential gene expression following exposure to cigarette smoke, which merit further investigation for clinical significance and as potential drug targets for future anticancer therapy development.

## 4. Materials and Methods

### 4.1. Animal Treatments

Male and female A/J mice were obtained from Jackson Laboratory (Bar Harbor, ME, USA) and housed in pathogen-free animal quarters at AeroCore Testing Services, University of Minnesota. All animal experiments were performed in accordance with the U.S. National Institutes of Health (NIH) Guide for the Care and Use of Laboratory Animals and was approved by the Institutional Animal Care and Use Committee, University of Minnesota.

Cigarette Smoke Exposure Conditions: Mice were placed in an exposure chamber connected to a TE-10B smoking machine (Teague Enterprises) adjusted to produce 89% side-stream and 11% mainstream smoke. The atmosphere within the exposure chamber was monitored to maintain the total particulate matter (TPM) at 100 mg/m^3^. During each exposure period, the TPM concentration for cigarette smoke-exposed groups was determined once per hour using a gravimetric filter–collection method. Carbon monoxide (CO) levels were monitored continuously in the smoke exposure by using an OM-EL-USB-CO USB data logger (Omega Engineering Inc., Norwalk, CT, USA).

Short-Term (3 weeks) Cigarette Smoke Exposure: Male and female A/J mice (6 weeks of age) were divided into two groups (14 male/14 female mice per group, with the sexes segregated). Mice in the cigarette smoke exposure group were exposed to cigarette smoke 4 h a day, 5 days a week, for three weeks. Mice in the control group were exposed to filtered air 4 h a day, 5 days a week, for three weeks. Mice were euthanized by CO_2_ asphyxiation in a chamber the day after their final cigarette smoke or control exposure. Lungs were harvested for Type II cell isolation as described below. Blood, brain, heart, and liver tissues were harvested and stored frozen at −80 °C for future analyses.

Long-Term (10 weeks) Cigarette Smoke Exposure: Six-week-old mice were divided into two groups as in the short-term exposure cohort. Mice in the cigarette smoke exposure group were exposed to cigarette smoke for 10 weeks in the same exposure conditions as the short-term exposure cohort. Mice in the control group were exposed to filtered air as in the short-term exposure cohort for 10 weeks. Mice were euthanized by CO_2_ asphyxiation in a chamber the day after their final cigarette smoke or control exposure. Lungs were harvested for Type II cell isolation as described below. Tissues were harvested and stored frozen at −80 °C as before.

Post-exposure Cigarette Smoke Treatment: Mice in this group were treated exactly as the animals in the long-term cigarette smoke exposure described above, but were allowed to recover in clean air for four weeks. Following post-exposure recovery, mice in this group were euthanized by CO_2_. Lungs were harvested for Type II cell isolation as described below. Tissues were harvested and stored frozen at −80 °C as before.

### 4.2. Histopathology Examination

A subset of mice from each exposure group (*n* = 4) was used for histological analysis. After the mice were sacrificed, the lungs were perfused, fixed in 10% buffered formalin, and embedded in paraffin. Sections were cut from paraffin-embedded tissues and stained with hematoxylin and eosin (H&E). Slides were analyzed for cellular inflammation under light microscopy by a pathologist at the UMN Comparative Pathology Shared Resource laboratory.

### 4.3. Isolation of Alveolar Type II Epithelial Cells

Alveolar type II epithelial cells (Type II pneumocytes) were isolated according to published procedures [68]. Briefly, the lungs were perfused with 10 mL cold phosphate buffered saline (PBS) before enzymatic digestion with 2 mL of dispase infused into the lung, after which they were removed and incubated in an additional 2 mL of dispase for one hour. The lungs were then manually disintegrated and the resulting cell suspension labeled with antibodies specific for CD11c, CD11b, F4/80, CD19, CD45, and CD16/CD32 (Thermo Fisher Scientific). Samples from 9 mice were pooled to include three sets of lungs per sample for a total of three samples for FACS separation. Type II pneumocytes were isolated by negative selection and thus identified as the unlabeled cell population [68]. Type II pneumocytes were also gated as sideward scatter high (SSC^high^) cell population which minimizes contamination with lymphoid cells by selecting cells with a higher granularity [68]. The cells were separated by fluorescence-activated cell sorting (FACS) by the University Flow Cytometry Resource at the University of Minnesota using a BD FACS Aria II P07800142 (BSL2) (BD Biosciences, San Jose, CA, USA).

### 4.4. Extraction of DNA, RNA, and Protein from Alveolar Type II Epithelial Cells

Following isolation via FACS, Type II pneumocytes were pelleted by centrifugation. The samples were first centrifuged for 12 min at 200× and 4 °C and the supernatant was removed, except for the bottom 1 mL, which was transferred to a 1.7 mL Eppendorf tube. To this tube, 500 µL of PBS was added, and the tube was centrifuged for 12 min at 200× *g* at 4 °C. The supernatant was removed while the bottom 100 μL was saved. One mL of PBS was added to the tube, and the sample was centrifuged for 12 min at 800× *g* and 4°C. After this final centrifugation, all supernatant is removed, and the cell pellet was saved for downstream analyses. A portion of the cell pellet (1.25 × 10^5^–5 × 10^5^ cells) was set aside to isolate protein. The remaining sample was used to isolate DNA and RNA using the Qiagen AllPrep DNA/RNA Mini Kit (Qiagen, Hilden, Germany) according to the manufacturer’s instructions. DNA and RNA were then quantified using Qubit fluorometric assay (Thermo Fisher Scientific, Fairlawn, NJ, USA).

### 4.5. RNA-Seq Analysis of Alveolar Type II Epithelial Cell RNA

After the extraction and quantification of RNA as described above, RNA integrity was confirmed using Agilent Bioanalyzer (Agilent, Santa Clara, CA, USA). RNA amounts for Type II pneumocytes pooled from 3 mice ranged between 300 ng and 4.1 µg. Total RNA samples were converted to Illumina sequencing libraries using SMARTer Stranded Total RNA-Seq Kit—Pico Mammalian Input (Takara Bio USA, Mountain View, CA, USA). RNA was reverse transcribed into cDNA and Illumina adapters were added using PCR. Sequencing of the cigarette smoke samples was performed on the Illumina NovaSeq 6000 sequencing system.

#### 4.5.1. RNA-Seq Read Processing

Raw reads were screened for low-quality bases and adapter contamination with FastQC version 0.11.7 (https://www.bioinformatics.babraham.ac.uk/projects/fastqc/, accessed on 15 May 2024). Raw reads were then cleaned of low-quality bases and adapter contamination with Trimmomatic version 0.33 [69]. Remaining reads were mapped to the human genome build GRCm38.p6 with HISAT2 [70]. Known exons and splice junctions were used to aid in read mapping. The resulting BAM files were filtered of reads with a mapping quality score of less than 60, keeping only uniquely mapped reads. These filtered BAM files were then sorted by read name and a counts matrix was generated with featureCounts from the subread version 1.6.2 package [71]. Quantification was performed at the gene level. Read pairs were assigned to a gene only if both mates mapped uniquely to the same gene, and if they mapped with the proper strand specificity. The scripts for performing the quality control, mapping, filtering, and generation of the counts matrix are available in a suite of scripts called CHURP [72].

#### 4.5.2. Gene Expression Quantification and Filtering

The counts matrix was analyzed with edgeR in the R statistical computing environment (R Core Team 2020) [73]. First, expression data for genes that were shorter than 200 bp were discarded. Then, an expression level filter was applied with the following procedure. The log2(CPM) value that corresponds to 10 fragments in the smallest library (hereafter *C*) was calculated. Then, the size of the smallest experimental group (hereafter *N*) was calculated. Genes that had a log2(CPM) value lower than *C* in *N* or more samples were removed, regardless of sample membership in any experimental group. One sample (3-week smoke exposure, sample MS2) was excluded because it had a very different gene expression profile from all other samples (Appendix A).

#### 4.5.3. Differential Gene Expression Testing

The filtered and normalized counts matrix was used to test a model that explains variation in gene expression as a function of treatment group, sex, age, and interactions among these three factors using the variancePartition package [74]. Based on the results of the variancePartition analysis, we tested differential gene expression between treatment groups using models that accounted for the effects of sex and explicitly tested for sex-specific responses (Appendix A). The differential gene expression models were built and analyzed in edgeR, and significance tests were performed with a quasi-likelihood F test. Genes were labeled as significantly different at a false discovery rate of 0.05 [73].

#### 4.5.4. Network Analysis

Gene expression values were also used to infer gene coexpression networks with the WGCNA package [75]. For input into WGCNA, raw counts were filtered in the same way as for differential gene expression testing. The filtered counts were normalized with the variance-stabilizing transformation implemented in DESeq2 [76]. A soft thresholding power of 11 was chosen for building the network because it was the lowest power where the scale-free topology model fit with an R^2^ value of at least 0.8. The network was built as an unsigned network and modules with a correlation of greater than 0.3 were merged. The eigengene values of the resulting modules were tested for association with treatment group, age, sex, and interactions among these factors with a one-way ANOVA. Modules were labeled as significantly associated with treatment group if their Bonferroni-adjusted *p*-values were less than 0.05. Scripts to perform gene expression analysis, variance partitioning analysis, differential gene expression testing, and coexpression analysis are available upon request.

### 4.6. RNA-Seq Validation via qRT-PCR

Following extraction and quantification of total RNA described above, its purity and integrity were confirmed using the Qubit 4 Fluorimeter (Invitrogen, Waltham, MA, USA). The first-strand complementary DNA was synthesized utilizing SuperScript IV VILO MasterMix (Invitrogen, Waltham, MA, USA) per manufacturer’s guidelines. Briefly, 8 ng of RNA was combined with ezDNAse digestion enzyme and diluted to 10 µL with 10× ezDNAse Buffer in RNAse-free water. Following incubation, 4 µL of MasterMix was added and the solution was diluted to a total working volume of 20 µL. A C1000 Touch Thermo Cycler (Bio-Rad, Hercules, CA, USA) was used to incubate the samples per the manufacturer’s recommendations for cDNA synthesis. The undiluted cDNA product was stored at −20 °C until use.

Quantitative reverse transcriptase-polymerase chain reaction (qRT-PCR) was carried out in 20 µL reaction volume on a StepOnePlus RT-PCR System (Applied Biosystems, Waltham, MA, USA) using SYBR Select Master Mix (Applied Biosystems, Waltham, MA, USA) and gene-specific primers at a final working concentration of 0.2 µM. For the PCR amplification, uracil DNA glycosylase (UDG) was activated at 50 °C for 2 min followed by AmpliTaq activation at 9°C for 2 min. Forty cycles of amplification were completed, consisting of denaturing at 95 °C for 15 s and a combined annealing/extension at 60 °C for 60 s. All samples were normalized to an internal control of β-actin. Comparative Ct of technical triplicates were used to assess the fold change in relative mRNA expression. Fold change differences were expressed for each test group relative to their respective control groups.

### 4.7. RRBS and Oxo-RRBS

Genomic DNA isolated from Type II alveolar epithelial cells of control and treated A/J mice was prepared for RRBS and oxo-RRBS using the Ovation RRBS Methyl-Seq System with TrueMethyl oxBS module (NuGEN, Redwood City, CA, USA) according to the manufacturer’s protocol. Library amplification was optimized as directed using qPCR, and the libraries were amplified accordingly followed by Agencourt bead clean-up. Libraries were quantified using the PicoGreen dsDNA assay (ThermoFisher, Waltham, MA, USA). Library size distribution was evaluated using the Bioanalyzer High-Sensitivity assay (Agilent). Paired-end sequencing (2 × 75 bp) was performed at the UMN Genomics center using an Illumina NextSeq 550 instrument (Illumina, San Diego, CA, USA) and the 150-cycle High-Output flow cell kit (Illumina, San Diego, CA, USA).

#### 4.7.1. Reduced Representation Bisulfite Sequencing Read Handling

Reduced representation bisulfite sequencing (RRBS) and oxidative RRBS (oxo-RRBS) reads were screened for low-quality bases and adapter contamination with FastQC version 0.11.7 (https://www.bioinformatics.babraham.ac.uk/projects/fastqc/, accessed on 15 May 2024). Sequencing reads were then trimmed of low-quality bases and adapter contaminants with TrimGalore! version 0.5.0_dev (https://www.bioinformatics.babraham.ac.uk/projects/trim_galore/, accessed on 15 May 2024), with custom adapter sequences for the NuGEN Ovation RRBS MethySeq kit. Cleaned reads were aligned to the GRCm38.p6 reference genome with Bismark version 0.19.0 [77]. Custom parameters were used to increase the sensitivity during read mapping. Resulting BAM files were then sorted, filtered of alignments with mapping quality lower than 20, and reading were mapped to “blacklisted” regions of the genome (ENCODE file ENCFF547MET).

#### 4.7.2. Methylation and Hydroxymethylation Analysis

Filtered BAM files from the RRBS and oxo-RRBS data were used to estimate methylation and hydroxymethylation levels with programs in the MethPipe version 3.4.3 suite of software [21]. BAM files were processed to generate estimated methylation levels at symmetric CpG sites in the mouse genome. Only sites covered by at least 10 sequence reads were retained for analysis. The hydroxymethylation level was estimated using the “mlml” program in the MethPipe suite [78]. Resulting methylation and hydroxymethylation level estimates were used to identify significantly differentially methylated regions (DMRs) and differentially hydroxymethylated regions (DhMRs). Individual CpG sites were tested for significance with the “radmeth” program in the MethPipe suite, and *p*-values were locally smoothed in 200 base pair windows with the “adjust” routine in the “radmeth” program. DMRs and DhMRs were retained as significant at the 0.05 false discovery rate and if they contained at least three significant CpG sites. DMRs and DhMRs that overlapped gene bodies and promoter regions were identified with bedtools version 2.29.0 [79]. Gene bodies and promoters that overlapped DMRs and DhMRs were used as input for a core analysis in Qiagen Ingenuity Pathway Analysis (IPA) to identify significantly enriched pathways.

### 4.8. DNA Digestion and HPLC Enrichment of mC and hmC

Genomic DNA extracted from alveolar Type II epithelial cells was used to quantify global levels of mC and hmC. DNA (0.5–2 µg) was spiked with internal standards (^13^C_10_^15^N_2_-5-methyl-2′-deoxycytidine and d_3_-5-hydroxymethyl- 2′-deoxycytidine). The DNA was then enzymatically digested to 2′-deoxynucleosides using PDE I (55 mU), PDE II (63 mU), DNase I (28 U), and alkaline phosphatase (48 U) in a buffer containing 10 mM Tris-HCl pH 7.0 and 15 mM magnesium chloride and then filtered through Nanosep 10K Omega filters (Pall Corporation, Port Washington, NY, USA).

DNA hydrolysates were dried and offline HPLC purification of mC, hmC, and their internal standards was achieved with an Agilent 1100 series HPLC system equipped with a UV detector and an automated fraction collector (Agilent Technologies, Santa Clara, CA, USA). An Atlantis T3 column (Waters, 4.6 × 150 mm, 3 µm) was eluted at a flow rate of 0.9 mL/min with a gradient of 5 mM ammonium formate buffer, pH 4.0 (A), and methanol (B). Solvent composition was changed linearly from 3% to 30% B over 15 min, increased to 80% over the next 3 min, then increased again to 96% over 1 min, and maintained at 96% B for 0.5 min. Solvent composition was returned to initial conditions (3% B) and the column was equilibrated for 7 min. dC was quantified by HPLC-UV using calibration curves obtained by analyzing authentic dC standards. HPLC fractions corresponding to mC and hmC were combined, dried, and analyzed by isotope dilution HPLC-ESI-MS/MS.

### 4.9. HPLC-ESI^+^-MS/MS Quantitation of Global Levels of mC and hmC

HPLC-ESI^+^-MS/MS quantitation of mC and hmC was performed using a Dionex Ultimate 3000 UHPLC (Thermo Fisher, Waltham, MA, USA) interfaced with a Thermo TSQ Quantiva mass spectrometer (Thermo Fisher). Chromatographic separation was achieved with a Zorbax SB-C18 column (0.5 × 150 mm, 5 µm, Agilent) eluted at a flow rate of 10 µL/min with a gradient of 2 mM ammonium formate (A) and acetonitrile (B). Solvent composition was changed linearly from 3% to 5% B over 6 min, increased to 43.5% B over the next 7 min, and maintained at 43.5% B for 1 min. Solvent composition was then returned to initial conditions (3% B) and the column was equilibrated for 7 min. Under these conditions, mC and ^13^C_10_^15^N_2_-mC eluted at 5.8 min, and both hmC and its internal standard (D_3_-hmC) eluted at 4.0 min. Quantitation was achieved by monitoring the transitions *m*/*z* 258.2 [M + H^+^] → *m*/*z* 142.1 [M–deoxyribose + H^+^] for hmC, *m*/*z* 261.2 [M + H^+^] → *m*/*z* 145.1 [M–deoxyribose + H^+^] for D_3_-hmC, *m*/*z* 242.1 [M + H^+^] → *m*/*z* 126.1 [M + H^+^] for mC, *m*/*z* 254.2 [M + H^+^] → *m*/*z* 133.1 [M + H^+^] for ^13^C_10_^15^N_2_-mC. Optimal mass spectrometry conditions were determined by infusion of authentic standards. Typical settings on the mass spectrometer were a spray voltage of 3000 V, a sheath gas of 15 units, the declustering voltage was 5 V, the RF lens was 40 V, the vaporizer temperature was 75 °C, and the ion transfer tube was maintained at 350 °C. The full width at half-maximum (FWHM) was maintained at 0.7 for both Q1 and Q3. MS/MS fragmentation was induced with collision gas pressure of 1.0 mTorr and a collision energy of 10.3 V.

### 4.10. Protein Extraction and Quantitation

To generate protein extract, designated cells for each sample were transferred to 0.45µm spin filters (Corning) and washed three times in cold PBS by suspending the cells in 500 µL buffer and centrifuging at 500× *g* to pellet the cells and pull the PBS through the filter, discarding the flowthrough. Following the washes, the cells where lysed via the application of 50 µL lysis buffer (100 mM TEAB pH = 8, 7 M urea, 2 M thiourea, 10% acetonitrile, and complete protease inhibitor tablets without EDTA) with vigorous pipetting, followed by centrifugation for 15,000 rpm for 15 min. The flow-through was collected and the protein concentration determined via Qubit Fluorometer (Thermo Scientific, Waltham, MA, USA), and stored at −80 °C until digestion

### 4.11. Protein Digestion and Processing

Recovered protein extracts were digested using single-pot solid-phase-enhanced sample preparation (SP3) beads [80]. Briefly, samples were reduced via the addition of DTT to 5 mM followed by incubation at 56 °C for 30 min. Following reduction, samples were alkylated via the addition of iodoacetamide to 8 mM and incubated in the dark at room temperature for 30 min. Following reduction and alkylation, protein samples were brought to a final volume of 48 µL with PBS, after which 2 µL of washed SpeedBead Magnetic Carboxylate bead mixture was added to each sample and the samples were mixed via pipetting. Next, ethanol was added to each sample to a final ethanol concentration of 70%, after which the samples were mixed again and allowed to settle on the benchtop. Samples were then added to a magnetic rack and the beads allowed to immobilize. The supernatant of each sample was removed and discarded, after which the pelleted beads in each sample were washed three times in 80% ethanol. The washed beads were sonicated for 1 min, after which the bead pellets in each sample were resuspended in 25 µL of 20 mM TEAB (pH 8.5) supplemented with trypsin at a concentration of 1:25 enzyme to approximate protein abundance. The samples were then incubated overnight at 37 °C to digest proteins immobilized on the beads. After the overnight digestion, samples were supplemented with an additional 25 µL of trypsin solution and digested for a further 2 h at 37 °C. Following the second digestion, sample beads were added immobilized on a magnetic rack and the supernatant removed and retained. To extract the remaining peptides, beads were resuspended in 50 µL of 0.1% formic acid and immobilized on a magnetic rack, with the supernatants removed and pooled with the first round of supernatants. Peptide samples were then quantified with 280 nm absorbance on the nanodrop, with 1µg aliquots set aside and dried down in the speed vac.

### 4.12. C18 Stage-Tip TMT Labeling of Peptides

A modified TMT-labeling strategy based on Myers et al. [81] was used to label digested peptides (Thermo Fisher Scientific, Waltham, MA, USA). Individual C18 spin columns were added to Eppendorf tubes for each sample to be processed, after which the columns were conditioned with methanol. Following conditioning, columns were further washed with 50 µL of 80% acetonitrile and 0.1% formic acid, and finally equilibrated with 50 µL of 0.1% formic acid. Dried-down 1 µg sample aliquots were resuspended in 50 µL 0.1% formic acid and run through the C18 columns twice at 1000 g for 1 min to immobilize peptides on the C18 columns. Following immobilization on the C18 columns, peptides were washed with two 50 µL aliquots of 0.1% formic acid. Prior to labeling, 0.8 mg TMT 11-plex reagent aliquots were brought to room temperature and were reconstituted in 41 µL of anhydrous acetonitrile to create TMT stock solutions. To label peptides, 50 µL of working TMT labeling solutions (1 µL TMT stock solution in 49 µL 20 mM TEAB, pH = 8) were added to each C18 spin column containing peptides and centrifuged for 2 min at 300 g. Labeling was repeated three more times, with each sample corresponding to a specific TMT channel. Following labeling, excess TMT reagents were washed away with two 50 µL washes of 0.1% formic acid. Samples were eluted from the C18 spin tips into new Eppendorf tubes with an initial addition of 50 µL 80% acetonitrile and 0.1% formic acid with subsequent centrifugation followed by a second elution with 50 µL of 80% acetonitrile in 20 mM ammonium formate pH = 10. Samples are then concatenated together and evaporated to dryness via speed vac.

### 4.13. High pH Fractionation of Peptides

To fractionate low amounts of peptide at high pH on reverse-phase stage tips, we used a protocol adapted from Dimayacyac-Esleta et al. [82] and Kim et al. [83]. Stage tips were prepared by packing single cutouts of C8 AttractSPE disk (Affinisep) into 200 µL pipette tips, after which each was loaded with 100 µL of 5 µm ReproSil-Pur C18-AQ (Dr. Maisch Gmbh, Ammerbuch DE) slurry (15µg/µL in 1:1 acetonitrile/100 mM ammonium formate, pH = 10) and further packed via 1500 g centrifugation for 2 min. Stage tips were then conditioned sequentially with the addition of 50 µL methanol, 80% acetonitrile in 100 mM ammonium formate, pH = 10, and 20% acetonitrile in 100 mM ammonium formate, pH = 10. Following conditioning, stage tips were transferred to fresh Eppendorf tubes with final equilibration of the columns carried out with 50 µL of 100 mM ammonium formate, pH = 10. Samples were reconstituted in 50 µL of 100 mM ammonium formate, pH = 10, and passed through stage tips twice with centrifugation at 1500× *g* for 2 min. With peptides immobilized on stage tips, the tips were transferred to fresh Eppendorf tubes and eluted with centrifugation at 1500× *g* for 2–3 min using multiple 50 µL aliquots of buffer containing 100 mM ammonium formate, pH = 10, and increasing amounts of acetonitrile. After elution, the 17 fractionations were concatenated into 9 fractions in LC-MS vials and evaporated to dryness via speed vac. For LC-MS analysis, samples were reconstituted in 10 µL of 0.1% formic acid in water.

### 4.14. HPLC-MS/MS Analysis of Proteins

Fractionated peptide samples were analyzed on an Orbitrap Fusion Tribrid Mass Spectrometer interfaced with an Ultimate 3000 UHPLC. The UHPLC was run in nanoflow mode with a reverse-phase nanoLC column (15 cm × 250 μm) packed with 5 μm diameter Luna C18 resin. Samples were reconstituted in 10 µL of buffer A (0.1% formic acid in water) prior to analysis. Samples were run on a 90 min gradient with a 5–22% increase in buffer B (0.1% formic acid in acetonitrile) over the first 71 min, followed by a 22–33% increase in buffer B over the next 5 min and rapid increase of 33–90% increase in buffer B over 5 min. Once the column was at 90% buffer B, the column was washed for 4 min, and finally a 90–95% decrease in buffer B over 2 min followed by a 3 min equilibration at 5% buffer B. Samples were run at a flow rate of 300 nL/min. Peptides were analyzed in positive mode using Top12 Full MS/dd-MS/MS mode with an expected chromatographic peak FWHM of 15 s. For the full MS scans, resolution was 70,000 with an AGC target of 1e6, a maximum IT of 30 ms, and a scan range of 300 to 2000 *m*/*z*. Tandem mass spectrometry (MS/MS) experiments were conducted at 30,000 resolutions, AGC target of 5e4, maximum IT of 50 ms, an isolation window of 2.0 *m*/*z*, and a normalized collision energy of 30. Data were collected in centroid mode.

### 4.15. Proteomics Data Analysis

Raw mass spectrometry data were analyzed together in MaxQuant [84] using the Reporter Ion MS/MS quantification mode against the Uniprot *Mus musculus* proteome supplemented with the contaminants database. Carbamidomethylation of cysteine was included as a fixed modification, while oxidation of methionine, N-terminal acetylation, and phosphorylation of serine, threonine, and tyrosine were included as variable modifications. Following analysis, data were analyzed using the open-source data manipulation platform Perseus [85] to generate volcano plots. Gene ontology (GO) analyses were conducted using gProfiler [86]. Proteomics data were compared with transcriptomic data generated from these same cells using QuanTP [87] within the Galaxy MSI suite. The proteomics raw files and MaxQuant results can be found in PRIDE with the accession PXD054404. 

## Figures and Tables

**Figure 1 ijms-25-09365-f001:**
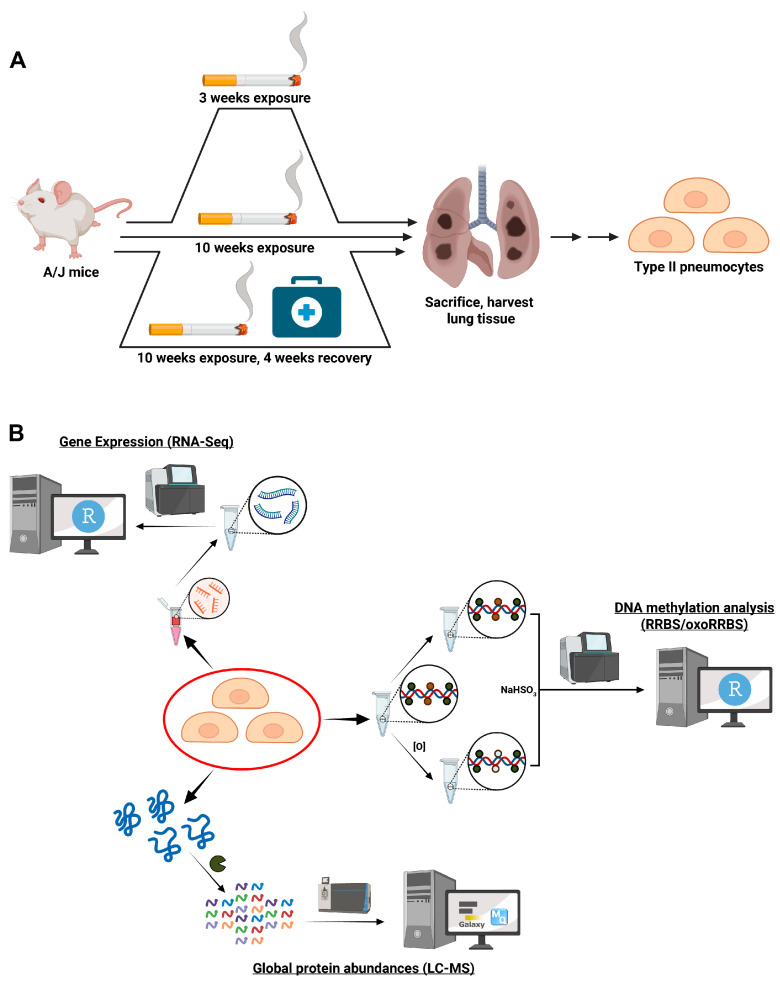
Animal Study Design. (**A**) A/J mice were treated with cigarette smoke. In the cigarette smoke exposure groups, the mice were exposed to cigarette smoke for 4 h per day, 5 days per week for either 3 or 10 weeks. Control mice were exposed to filtered air for the same amount of time. A third group received the 10-week CS treatment and was allowed to recover in air for a further 4 weeks. (**B**) Multi-omic strategy for the analysis of Type II pneumocytes. Figures generated via biorender.com.

**Figure 2 ijms-25-09365-f002:**
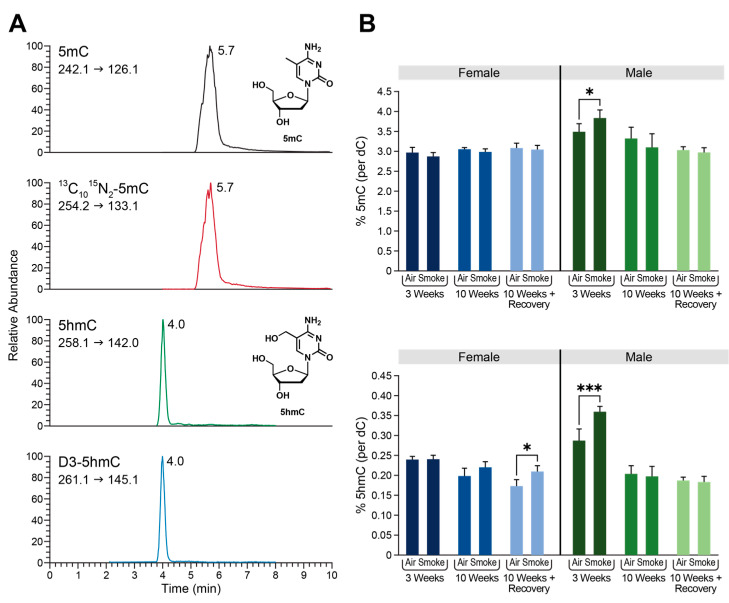
Global changes in DNA methylation and hydroxymethylation in Type II alveolar cells of mice exposed to ECS. (**A**) Representative LCMS trace for global quantitation of 5 mC/5 hmC. From top to bottom: 5 mC, ^13^C_10_^15^N_2_-5-methyl-2′-deoxycytidine, 5 hmC, 5-hydroxymethyl-d_2_-2′-deoxycytidine-6-d_1_. (**B**) Global levels of 5mC/5hmC in Type II cells of A/J mice exposed to ECS. Data are expressed as percentage of dC and represents mean values ± SD of at least three animals. The specific treatment and duration are shown on the x-axis. * *p* < 0.05, *** *p* < 0.001.

**Figure 3 ijms-25-09365-f003:**
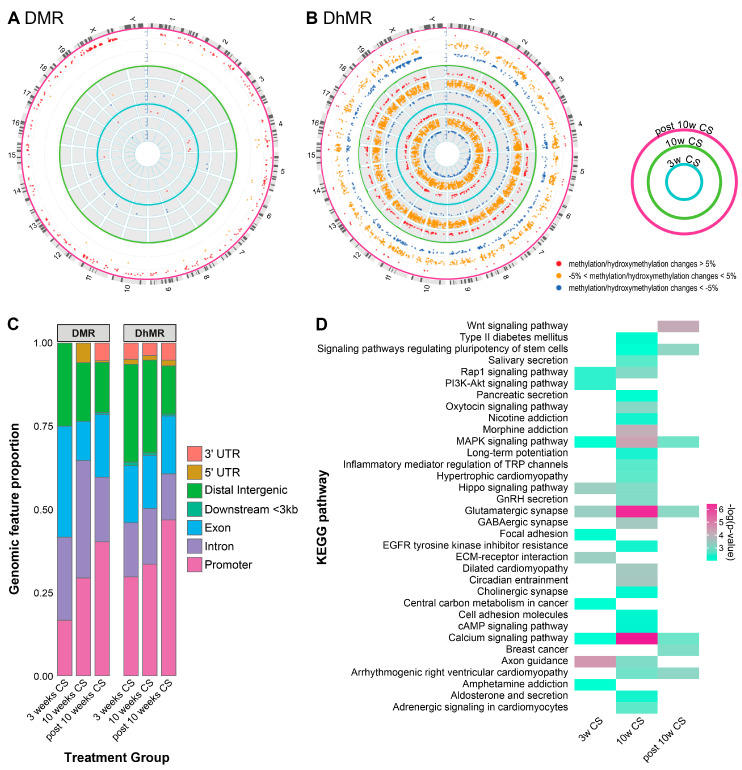
Overview of RRBS/oxRRBS profiling in all three treatment groups. (**A**,**B**) Circos plot showing DMRs (**A**) and DhMRs (**B**) plotted across chromosomes in Type II pneumocytes of female A/J mice exposed to ECS, and the corresponding controls. Circos plot legend: from the outer circle to inner circle: 3 weeks of smoking, 10 weeks of smoking, post-exposure treatment. Red dots represent hypermethylation/hydroxymethylation region and blue dots represent hypomethylation/hydroxymethylation. (**C**) Stacked percentage bar plot showing genomic feature annotations of DMR/DhMR identified in all 3 treatment groups. DMR/DhMR is defined as a genomic region that contains at least 3 CpG sites within a 200 bp genomic window with a false discovery rate of less than 0.05. (**D**) Heatmap to show KEGG pathways enriched in each treatment group via over representation analysis of genes containing DhMRs. Grey tiles indicate no significant enrichment.

**Figure 4 ijms-25-09365-f004:**
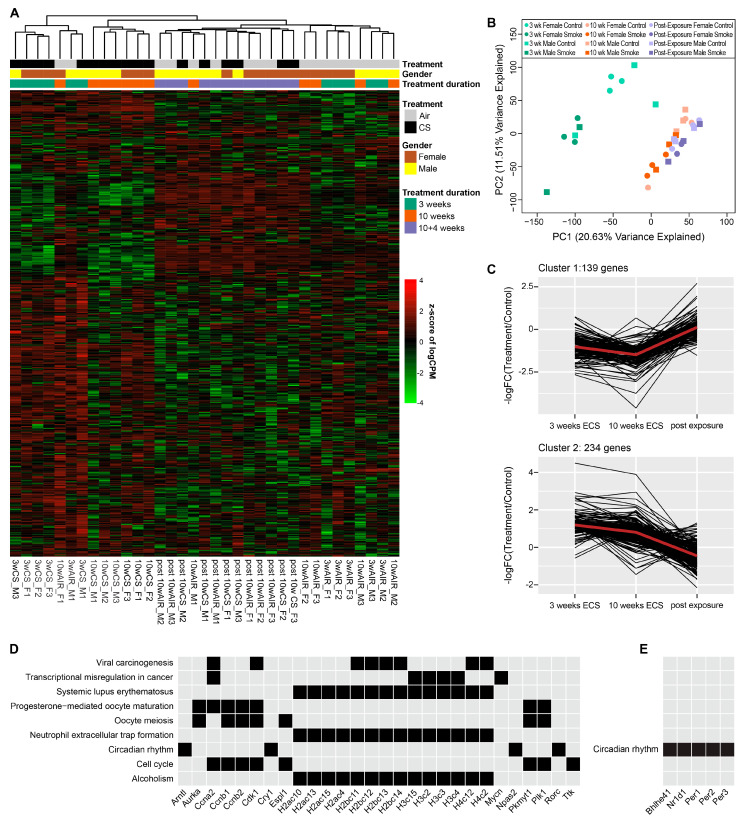
Overview of the RNA-seq results for Type II alveolar cells from mice exposed to ECS. (**A**) Heatmap of the top 500 variant genes across all samples. The plotted values are z-scores of logCPM values across all samples. (**B**) Principle component analysis of the gene expression changes observed in type II cells following exposure to cigarette smoke for 3 weeks, 10 weeks, or 10 weeks with 4 weeks post-exposure. Samples separate into distinct clusters based mainly on age of the mice. (**C**) Clustering analysis of the DEGs detected in all treatment groups. (**D**,**E**) KEGG pathway analysis of genes in Cluster1 (**D**) and Cluster2 (**E**).

**Figure 5 ijms-25-09365-f005:**
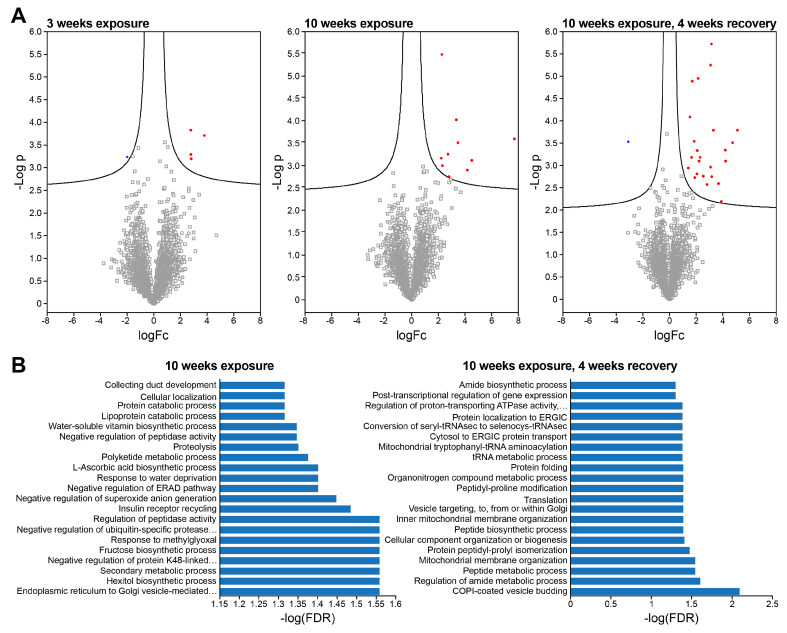
Protein abundances show significant changes in response to ECS exposure in Type II pneumocytes of female A/J mice. (**A**) Differential protein abundances after 3 weeks of exposure, 10 weeks of exposure, and 10 weeks of exposure with 4 weeks of recovery. Statistically significant increases are marked in red, decreases are marked in blue. (**B**) Gene ontology analysis results for protein changes in Type II pneumocytes of A/J mice observed immediately following 10-week exposure to ECS (**left**) and after 4 weeks of recovery.

**Figure 6 ijms-25-09365-f006:**
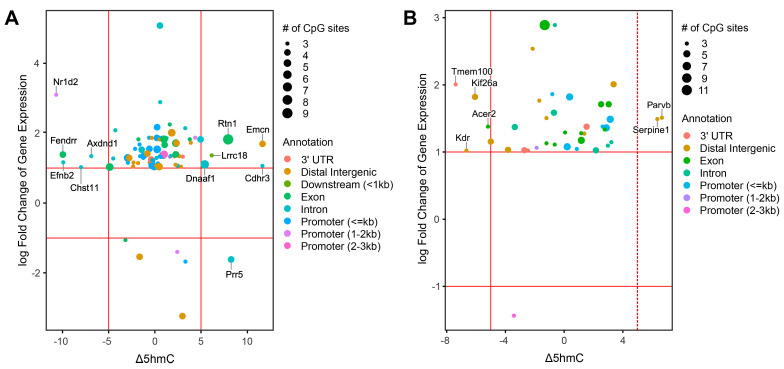
Integration of ECS-induced epigenomic and transcriptomic changes identifies DEGs regulated by DNA methylation and hydroxymethylation. (**A**) Integration of DhMR and DEG in Type II cells of female mice following 3 weeks of exposure to ECS. Only genes with DMR difference greater than 5% are labeled. (**B**) Integration of DhMR and DEG in 10 weeks of ECS exposure. Only genes with DhMR difference greater than 5% are labeled.

**Figure 7 ijms-25-09365-f007:**
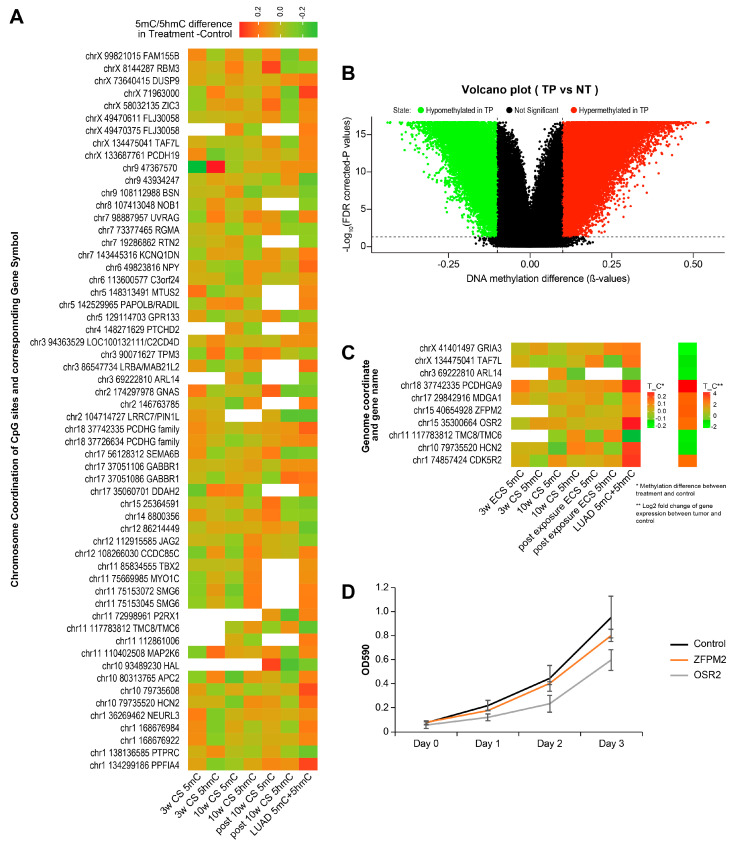
Comparison of smoking-induced epigenetic/transcriptomic changes in A/J mouse model observed in this study to the epigenetic/transcriptomic dysregulation of human lung adenocarcinoma (LUAD). The results shown here are in whole or part-based upon data generated by the TCGA Research Network: https://www.cancer.gov/tcga (accessed on 31 August 2021 (RNA-seq dataset) and on 24 November 2021 (DNA methylation dataset)). (**A**) A volcano plot of differentially methylated sites in human lung adenocarcinoma. TP: primary tumor, NT: normal tissue. DMS is significant when adjusted *p* value < 0.05 and methylation difference greater than 10%. (**B**) A heatmap of 58 DMS that are shared between mouse to ECS exposure and human LUAD. Y-axis showed the mouse genome coordination of the DMS and corresponding gene name. (**C**) A heat map of 10 genes that showed early epigenetic change upon ECS exposure, which persist in human LUAD and are associated with differential expression of the corresponding genes. (**D**) Knockdown of OSR2 significantly inhibited H838 cells’ proliferation as compared to the non-targeting siRNA group.

## Data Availability

The sequencing data have been deposited to NCBI GEO under accession GSE275545.

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
