# Peer review of "A Multi-Omics Study of Epigenetic Changes in Type II Alveolar Cells of A/J Mice Exposed to Environmental Tobacco Smoke"

_ijms, 2024, doi:10.3390/ijms25179365_

Round 1

Reviewer 1 Report

Comments and Suggestions for Authors

The manuscript is well-written and tackles an intriguing topic, but it requires several improvements before it can be considered for publication.

  1. The introduction needs a more thorough and detailed description of the known mechanisms underpinning the hypothesis. 

  2. In line 54, the authors begin discussing results, which is inappropriate for the introduction section. This information should be reserved for the results section to maintain a clear and logical structure.

  3. The results presented in section 2.1 (Animal Studies) should be summarized in a table for improved clarity and readability.

  4. References in superscript should be placed at the end of sentences rather than in the middle, as seen in lines 206 and 208. 

  5. The manuscript does not clearly explain the rationale behind the selection of the specific time intervals of 3, 10 (exposure), and 4 weeks (recovery) used in the experiment. Considering that carcinomas develop over a long period due to the accumulation of genetic and epigenetic abnormalities, it is essential to discuss whether the chosen experimental durations (3, 10, and 4 weeks) are sufficient to draw relevant and meaningful conclusions.
Comments on the Quality of English Language

Moderate editing of English language required.

Author Response

Response to Reviewers' comments:

Reviewer 1

  1. The introduction needs a more thorough and detailed description of the known mechanisms underpinning the hypothesis.

Response: We thank the reviewer for their suggestion and have added more reviews of the known mechanisms underpinning the hypothesis.

  1. In line 54, the authors begin discussing results, which is inappropriate for the introduction section. This information should be reserved for the results section to maintain a clear and logical structure.

Response: We thank the reviewer for their suggestion and have removed the summary of our findings in the introduction section.

  1. The results presented in section 2.1 (Animal Studies) should be summarized in a table for improved clarity and readability. – may be it is better to have a bar graph with p-value? Thoughts?

Response: We thank the reviewer for their suggestion and have summarized the body weight changes in table S-1.

Table S-1 Body weight of mice with 10 weeks air or ECS exposure.

Gender

Filtered Air Control

ECS

Male

24.68 ± 2.49 g

21.34 ± 1.37 g

Female

19.41± 1.19 g

16.86 ± 0.62 g

  1. References in superscript should be placed at the end of sentences rather than in the middle, as seen in lines 206 and 208.

Response: We thank the reviewer for this comment. However, in text superscript reference is not uncommon as the reference in lines 206 and 208 only apply to half of the sentence. For example, this paper “Saint-André, V., Charbit, B., Biton, A. et al. Smoking changes adaptive immunity with persistent effects. Nature 626, 827–835 (2024). https://doi.org/10.1038/s41586-023-06968-8” has quite a few in text superscript references.

  1. The manuscript does not clearly explain the rationale behind the selection of the specific time intervals of 3, 10 (exposure), and 4 weeks (recovery) used in the experiment. Considering that carcinomas develop over a long period due to the accumulation of genetic and epigenetic abnormalities, it is essential to discuss whether the chosen experimental durations (3, 10, and 4 weeks) are sufficient to draw relevant and meaningful conclusions.

Response: We thank the reviewer for their comment. The timing of this study was chosen to specifically address our research question, which focuses on the early epigenetic dysregulation that would trigger carcinogenesis. 3 weeks was chosen as it is a relatively short term treatment. 10 weeks was chosen as a relatively long term treatment but also with a good survival rate for the mice during the study. The selected timeframe allows us to observe the progression and reversibility of the epigenetic dysregulation before tumor formation, which might be the trigger of smoking associated lung cancer.

Reviewer 2 Report

Comments and Suggestions for Authors

Here, the authors investigate the genetic, epigenetic and proteomics changes in Type II alveolar cells of A/J mice exposed to environmental tobacco smoke. The researchers found significant changes in DNA methylation, gene expression, and protein abundance that contribute to an inflammatory and potentially oncogenic phenotype. The study highlights the role of inflammation in the carcinogenicity of cigarette smoke and identifies specific epigenetic markers and pathways involved in lung cancer development. This article is well-written, and the figures are of high quality/ the statistical analysis appropriate.

Major Points:

Here, the authors use a multi-omics approach, integrating epigenomics, transcriptomics, and proteomics. The integration of these datasets enable a comprehensive understanding of the changes observed at the molecular level in the ATII cells upon smoking. 

The findings provide novel insights into the mechanisms of smoking-induced lung cancer, which is a major public health issue.

Minor Points:

The study has a limited Scope as it focuses only on Type II alveolar cells, which may not capture the full complexity of lung cancer development. Some changes could also be observed in ATI or basal cells. This should be discussed in a limitation paragraph at the end of the article. 

High mortality rate in the 10-week exposure group could affect the study’s outcomes. The mortality is caused by changes in the immune infiltrate, and/ or vascular damage (Table S1). This should also be discussed further.

Please also submit the raw data for the different OMICs experiments to the appropriate databases (GEO, PRIDE) and add the reference number to the uploaded dataset to the methodology.

Author contribution is incomplete.

Author Response

Response to Reviewers' comments:

Reviewer 2

1.The study has a limited Scope as it focuses only on Type II alveolar cells, which may not capture the full complexity of lung cancer development. Some changes could also be observed in ATI or basal cells. This should be discussed in a limitation paragraph at the end of the article.

Response: We thank the reviewer for this comment and have discussed this limitation in the paragraph of the study limitation:” Furthermore, this study has only investigated the omics changes in type II alveolar epithelial cells, whereas other types of cells such as pulmonary neuroendocrine cells, basal cells and club cells could also be the origin of lung cancer or indirectly contribute to carcinogenesis through cell signaling transduction1; future studies could harness single cell technology to capture the full complexity of lung cancer development.”

  1. High mortality rate in the 10-week exposure group could affect the study’s outcomes. The mortality is caused by changes in the immune infiltrate, and/ or vascular damage (Table S1). This should also be discussed further.

Response: We thank the reviewer for this comment and have discussed this in the paragraph of the study limitation. We agree that the vascular congestion and intra-aveolar hemorrhage might cause confounding effects besides inflammation associated omics changes. Future studies should consider longer exposures with reduced dosage on each day.

  1. Please also submit the raw data for the different OMICs experiments to the appropriate databases (GEO, PRIDE) and add the reference number to the uploaded dataset to the methodology.

Response: We thank the reviewer for their comment. We have uploaded the RNA-seq and RRBS/oxRRBS dataset to GEO (GEO accession number: GSE275545) and proteomics dataset to PRIDE (accession number: PXD054404). Their reference numbers have also been added to the corresponding methodology section.

  1. Author contribution is incomplete.

Response: We thank the reviewer for their comment. The authors contribution list has been updated in the corresponding section.

Reviewer 3 Report

Comments and Suggestions for Authors

The authors conducted multi-omic analyses on Type II pneumocytes from A/J mice exposed to cigarette smoke to explore lung cancer in smokers with inflammatory. The results showed significant changes in DNA methylation, gene expression, and protein abundance, which were partially reversible and contributed to an inflammatory and potentially cancerous state. 

- I highly suggest to add "multi-omics analysis" to the keywords.

-I highly suggest survival analysis using Kaplan-Meier or Cox-regression.

-I suggest adding a conclusion and/or discussion section that has the limitation, future work, and the importance of the findings from the medical application point of view.

Comments on the Quality of English Language

One time going through the paragraphs and the flow of text for better readability.

Author Response

Response to Reviewers' comments:

Reviewer 3

The authors conducted multi-omic analyses on Type II pneumocytes from A/J mice exposed to cigarette smoke to explore lung cancer in smokers with inflammatory. The results showed significant changes in DNA methylation, gene expression, and protein abundance, which were partially reversible and contributed to an inflammatory and potentially cancerous state.

- I highly suggest to add "multi-omics analysis" to the keywords.

Response: We thank the reviewer for their suggestion and have updated the keywords to reflect their wishes.

-I highly suggest survival analysis using Kaplan-Meier or Cox-regression.

Response: We thank the reviewer for their suggestion and have included a Kaplan-Meier plot in our supplemental information, making reference to it on lines 76 and 77 of the manuscript.

-I suggest adding a conclusion and/or discussion section that has the limitation, future work, and the importance of the findings from the medical application point of view.

Response: We appreciate the reviewer's concerns and have added a paragraph to the discussion circa line 517 discussing some of the shortfalls of this study that future experiments will endeavor to rectify.